# Comparative Transcriptomic Analyses Reveal the Regulatory Mechanism of Nutrient Limitation-Induced Sporulation of *Antrodia cinnamomea* in Submerged Fermentation

**DOI:** 10.3390/foods11172715

**Published:** 2022-09-05

**Authors:** Huaxiang Li, Dan Ji, Zhishan Luo, Yilin Ren, Zhenming Lu, Zhenquan Yang, Zhenghong Xu

**Affiliations:** 1College of Food Science and Engineering, Yangzhou University, Yangzhou 225009, China; 2School of Biotechnology, Jiangnan University, Wuxi 214122, China; 3Department of Gastroenterology, Affiliated Hospital of Jiangnan University, Wuxi 214041, China; 4Jiangsu Key Laboratory of Dairy Biotechnology and Safety Control, Yangzhou University, Yangzhou 225009, China

**Keywords:** *Antrodia cinnamomea*, transcriptomic, submerged fermentation, asexual sporulation, nutrient limitation, regulatory mechanism

## Abstract

*Antrodia cinnamomea* is a precious edible and medicinal mushroom with various biological activities, such as hepatoprotection, antitumor, antivirus, immunoregulation, and intestinal flora regulation. However, the wild fruiting bodies of *A. cinnamomea* are scarce and expensive. Submerged fermentation based on spore inoculation has become the most efficient and popular artificial culture method for *A. cinnamomea*. In order to complement the mechanism of asexual sporulation of *A. cinnamomea* in submerged fermentation, and provide a theoretical basis to further improve the sporulation, comparative transcriptomics analysis using RNA-seq and RT-qPCR were conducted on *A. cinnamomea* mycelia cultured under different nutritional conditions to reveal the regulatory mechanism underlying the asexual sporulation induced by nutrient limitation. The obtained mechanism is as follows: under nitrogen starvation, the corresponding sensors transmit signals to genes, such as *areA* and *tmpA*, and promote their expression. Among these genes, AreA has a direct or indirect effect on *flbD* and promotes its expression, further enhancing the expression of *brlA*. Meanwhile, TmpA has a direct or indirect effect on *brlA* and promotes its expression; under carbon starvation, transport protein Rco-3, as a glucose sensor, directly or indirectly transmits signals to *brlA* and promotes its expression. BrlA promotes the expression of *abaA* gene, which further enhances the expression of *wetA* gene, and *wetA* then directly leads to asexual sporulation and promotes spore maturation; meanwhile, *gulC* can also promote cell autolysis, which provides energy and raw materials for sporulation.

## 1. Introduction

*Antrodia cinnamomea* (syn. *Antrodia camphorata*) belongs to phylum Basidiomycetes, family Polyporaceae, and genus *Antrodia* [1]. It is a precious edible and medicinal mushroom with various biological activities, such as hepatoprotection, antitumor, hypoglycemic, antioxidant, antiviral, immunoregulation, and gut microbiota regulation [2,3,4,5]. Its main active substances include triterpenoids and polysaccharides [6].

*A. cinnamomea* has a huge market demand due to its outstanding biological activity and medicinal value. However, the wild fruiting bodies of *A. cinnamomea* with a single host grow slowly. Therefore, the demand exceeds the supply, and the commodity becomes extremely expensive. To explore alternative resources for wild *A. cinnamomea* fruiting bodies, numerous studies focused on the large-scale artificial culture of *A. cinnamomea*. At present, the four main techniques for the artificial culture of *A. cinnamomea* are basswood cultivation, plate culture, solid-state fermentation, and submerged fermentation [7]. Characterized by short fermentation period, high production efficiency, and easy to large-scale application, submerged fermentation has become the most popular artificial culture method for *A. cinnamomea* [8].

Mycelium inoculation is usually used in the submerged fermentation of *A. cinnamomea* [9,10,11]. However, the seed (inoculum) quality and amount of mycelia are not easy to control, and the poor synchronization of mycelium growth during fermentation results in poor batch stability. Asexual spore inoculation can be adopted to control the inoculation amount and seed quality, and improves the controllability and batch stability of the fermentation [12]. Therefore, asexual spore inoculation has become the development trend of *A. cinnamomea* submerged fermentation. In our previous studies, rapid submerged fermentation and repeated submerged fermentation based on asexual spore inoculation were established, and these processes greatly improved the production efficiency of the active substances of *A. cinnamomea* [8,12].

Current studies on the submerged fermentation of *A. cinnamomea* usually focused on increasing the biomass and active substance production of *A. cinnamomea* by optimizing medium composition, environmental factors, and the fermentation process, or by adding precursor substances [11,13,14]. However, there are still some problems in the large-scale production of *A. cinnamomea* in submerged fermentation, such as the preparation for inoculum (i.e., the spores of *A. cinnamomea*) is tedious and time-consuming, which severely limits the production efficiency and benefit for *A. cinnamomea* in submerged fermentation. Revealing the regulatory mechanism is helpful to rationally adjust and control the asexual sporulation and thereby improve the efficiency of preparation for inoculum during submerged fermentation of *A. cinnamomea*. In our previous research, the molecular regulatory mechanism mediated by FluG in the asexual sporulation of *A. cinnamomea* submerged fermentation was revealed [15]. The present work aims to further study the sporulation of *A. cinnamomea* under nutrition deficiency in the late stage of submerged fermentation, and explore the molecular regulatory mechanism underlying asexual sporulation induced by nutrient limitation in submerged fermentation of *A. cinnamomea*.

## 2. Materials and Methods

### 2.1. Materials

*A. cinnamomea* strains (No. ATCC 200183) were purchased from the American Type Culture Collection (Manassas, VA, USA). agarose, goldview dye, PCR mix, and primer were obtained from Sangon Biotech (Shanghai, China). SYBR Select Master Mix for CFX was acquired from Life technologies (Carlsbad, CA, USA), and yeast extract was purchased from Oxoid Limited (Basingstoke, Hants, UK). The components of the culture medium and other conventional reagents were bought from Sinopharm Chemical Reagent Co., Ltd. (Shanghai, China).

### 2.2. Medium

The seed culture medium comprised the following: 20.0 g/L glucose, 1.0 g/L yeast extract powder, 3.0 g/L KH_2_PO_4_, 1.5 g/L MgSO_4_, and initial pH of 5.0 [15];

The fermentation medium comprised the following: 20.0 g/L glucose, 100 g/L yeast extract powder, 3.0 g/L KH_2_PO_4_, 1.5 g/L MgSO_4_, and initial pH of 3.9 [15];The basal medium comprised the following: 15.0 g/L glucose, 5.0 g/L yeast extract powder, 1.44 g/L KH_2_PO_4_, 0.38 g/L MgSO_4_, and initial pH of 3.9.

### 2.3. Seed Preparation and Submerged Fermentation of A. cinnamomea

*A. cinnamomea* mycelia on the slope of PDA were transferred to a 500 mL conical flask with 100 mL of seed culture medium. After 13–15 days of culture at 26 °C and 150 r/min, the fermentation broth was filtered with four layers of gauze. The obtained filtrate was *A. cinnamomea* inoculum (i.e., spore suspension). The spores in the spore suspension of *A. cinnamomea* were counted using a hemocytometer under an optical microscope, the spore concentration was then calculated. Afterward, the spore suspension was inoculated in a 500 mL conical flask with 100 mL of fermented medium at 1.0 × 10^6^ spores/mL and cultured at 26 °C and 150 r/min for 10 days [15].

### 2.4. Detection Methods

Detection of fermentation broth components: the fermentation broth cultured for 5 days was filtered with four layers of gauze to collect the filtrate. The glucose content was detected by a biosensor analyzer (SBA-40C, Shandong academy of sciences, China), the nitrogen content was detected by an element analyzer (VARIOEL III, Elementar Analysensyetem GmbH, Langenselbold, Germany), and the SO_4_^2−^ and PO_4_^3−^ content was detected by an atomic absorption spectrophotometer (DW-AA2081, Spectro, Kleve, Germany);Detection of contents of carbon and nitrogen in yeast extract powder: the element analyzer (VARIOEL III, Elementar, Langenselbold, Germany) was used for detection;Detection of the pH of fermentation broth: the pH meter (MettLer-ToLedo, Columbus, OH, USA) was used for detection [8];Detection of biomass: the fermentation broth was filtered with four layers of gauze. After the obtained mycelium pellets were washed with deionized water three times, they were dried at 75 °C to constant weight, weighted, and calculated for their biomass [8];Detection of sporulation: the spores were counted using a hemocytometer under an optical microscope, and the sporulation was then calculated [8].

### 2.5. Effects of Different Nutritional Conditions on Sporulation of A. cinnamomea

#### 2.5.1. Effects of Different Contents of Yeast Extract Powder on Sporulation of *A. cinnamomea*

*A. cinnamomea* spores were inoculated in 100 mL of fermented medium at 1.0 × 10^6^ spores/mL, cultured at 26 °C and 150 r/min for 5 days and allowed to stand for 5 min. The supernatant was then removed with pipette under sterile conditions. The mycelium pellets were washed with normal saline three times, then added with 100 mL of medium containing different contents of yeast extract powder (0, 0.5, 1.0, 5.0, or 10.0 g/L yeast extract powder, 15.0 g/L glucose, 1.44 g/L KH_2_PO_4_, 0.38 g/L MgSO_4_, initial pH of 3.9), and continuously cultured for 4 days. Then, sporulation and biomass were detected.

#### 2.5.2. Effects of Different Contents of Glucose on Sporulation of *A. cinnamomea*

*A. cinnamomea* spores were inoculated in 100 mL of fermented medium at 1.0 × 10^6^ spores/mL, cultured at 26 °C, and 150 r/min for 5 days, and allowed to stand for 5 min. The supernatant was then removed with a pipette under sterile conditions. The mycelium pellets were washed with normal saline three times, added with 100 mL of medium containing different contents of glucose (0 g/L yeast extract powder, 0, 2.0, 5.0, 10.0, or 20.0 g/L glucose, 1.44 g/L KH_2_PO_4_, 0.38 g/L MgSO_4_, initial pH of 3.9), and continuously cultured for 4 days. Sporulation and biomass were then detected.

### 2.6. Sample Preparation and RNA Extraction of A. cinnamomea Mycelia

#### 2.6.1. Sample Preparation of Mycelia

The fermentation broth of *A. cinnamomea* cultured in basal medium, nutrient-limited medium (glucose 5.0 g/L, yeast extract powder 0 g/L, the rest same as the basal medium) and nutrient-rich medium (glucose 15.0 g/L, yeast extract powder 10.0 g/L, the rest same as the basal medium) in the last 4 days was filtered with four layers of gauze. *A. cinnamomea* mycelium pellets were collected, washed with Tris-EDTA buffer solution (pH 8.0) five times, and stored in a refrigerator at −80 °C after snap-frozen using liquid nitrogen.

#### 2.6.2. Total RNA Extraction

Total RNA of the above *A. cinnamomea* mycelium pellets were extracted using UNIQ-10 column Trizol total RNA extraction kit (Sangon Biotech, Shanghai, China). The concentration and quality of the total RNA were detected by NanoDrop 2000c spectrophotometer and agarose gel electrophoresis, respectively. The RNA samples with final concentration ≥ 250 ng/µL, 1.9 ≤ A260/A280 ≤ 2.1, OD260/230 ≥ 1.8, and clearly showing three bands on gel were selected for RNA-seq and RT-qPCR.

### 2.7. RNA-Seq and Bioinformatic Analysis

The qualified total RNA samples were sent to Beijing Novogene Biotechnology Co., Ltd. (Beijing, China) for the high-throughput sequencing by Illumina HiSeq™ 2500. The low-quality read fragments obtained from sequencing were removed by Trim Galore software, and the qualified reads were subjected to de novo assembly by Trinity software to obtain the transcript database of *A. cinnamomea*. After the transcripts were interpreted and analyzed using Transdecoder software and compared against the genome database of *A. cinnamomea* (accession number: GCA_000766995.1, NCBI), the unigene database (Appendix A) of *A. cinnamomea* was obtained. Then, the principal component analysis and correlation analysis (with a Pearson correlation coefficient) were performed based on the FPKM value of unigene to check the repeatability of the samples.

After the expression of unigenes was analyzed and compared among different samples, the differentially expressed genes (or proteins) were obtained. Annotation and functional analysis were conducted on the differentially expressed genes through the NCBI database (https://www.ncbi.nlm.nih.gov/ accessed on 27 November 2021), GO database (http://geneontology.org/ accessed on 27 November 2021), and KEGG database (http://www.genome.jp/kegg/ accessed on 16 December 2021). With information from relevant references as the basis, the genes that may be related to sporulation induced by nutrient limitation in *A. cinnamomea* were finally obtained.

### 2.8. RT-qPCR Analysis

The qualified total RNA sample was diluted to the concentration of 50 ng/μL and reversely transcribed using M-MuLV first-strand cDNA synthesis kit (Sangon Biotech, Shanghai, China). RT-qPCR analysis was performed by taking the reversely transcribed cDNA as a template. The RT-qPCR reaction system is as follows: 8.5 μL of SYBR dye, 0.425 μL of forward primer, 0.425 μL of reverse primer, 1.25 μL of cDNA template, and 6.4 μL of ddH_2_O. The RT-qPCR reaction procedure is as follows: pre-denaturation at 95 °C for 10 min, followed by 45 cycles at 95 °C for 15 s, and at 60 °C for 60 s. The 2^−ΔΔCt^ calculation method [16] was applied to quantify the transcription level of genes using the 18S rRNA sequence of *A. cinnamomea* as the internal reference and fermented *A. cinnamomea* mycelium sample as the control. The RT-qPCR primer sequences of related genes are shown in Table 1.

### 2.9. Statistical Analysis of Data

At least three replicates were prepared for each experimental group. Data were presented as mean ± SD. One-way ANOVA was carried out using SPASS PASW Statistics Version 18.0 (CA, USA). Significant difference level was set as *p* < 0.05.

## 3. Results and Discussion

### 3.1. Components of Fermentation Broth Cultured for 5 Days

Rich nutrition is conducive to spore germination, and deficient nutrition is conducive to spore production [17]. Thus, the effect of nutritional conditions on the asexual sporulation of *A. cinnamomea* in submerged fermentation was studied to minimize the error caused by the difference in the growth status of mycelia. First, *A. cinnamomea* spores were inoculated in the medium with rich nutrition and cultured for 5 days (at this time, spores have not yet been produced and the mycelium pellets are growing well [8]. The mycelium pellets were continuously cultured for 4 days after changing the medium with different nutritional contents. Then, the effect of nutritional conditions on the asexual sporulation of *A. cinnamomea* in submerged fermentation was investigated.

The main components and pH of the fresh medium and 5-day cultured fermentation broth of *A. cinnamomea* were first tested (Table 2). Moreover, the contents of carbon (C) and nitrogen (N) in the organic nitrogen source of yeast extract powder used in the medium were calculated. The result shows that the contents of carbon and nitrogen in the yeast extract powder are 39.60% and 11.27%, respectively. Table 2 indicates that the carbon in the medium or fermentation broth mainly comes from glucose, the nitrogen mainly comes from yeast extract powder, the PO_4_^3−^ mainly comes from KH_2_PO_4_, and the SO_4_^2−^ mainly comes from MgSO_4_. Therefore, the components of the medium can be calculated according to the contents of nitrogen, SO_4_^2−^, PO_4_^3−^, and glucose in the fermentation broth. As listed in Table 2, the fermentation broth cultured for 5 days has a pH 3.9 and consists of (g/L): glucose 15.0, yeast extract powder 5.0, KH_2_PO_4_ 1.44, and anhydrous MgSO_4_ 0.38. These parameters are considered as the basic medium conditions.

### 3.2. Effects of Different Nutritional Conditions on the Asexual Sporulation of A. cinnamomea

After the *A. cinnamomea* fermentation broth was fermented for 5 days in the fermentation medium, the mycelium pellets were isolated and continuously cultured in 100 mL of medium containing different contents of nitrogen (yeast extract powder) or carbon (glucose) for 4 days. Then, sporulation and biomass were detected (Table 3).

The following results are presented in Table 3: (1) No significant difference in biomass, sporulation capacity, and residual sugar content is observed between “CK1” and “CK2” groups, indicating that normal saline washing has no significant effect on the growth and sporulation of *A. cinnamomea*. (2) The change in concentration of yeast extract powder (nitrogen source) has no significant effect on the biomass of *A. cinnamomea* but has an extremely significant effect on its sporulation capability. If the yeast extract powder is deficient, then it is conducive to the sporulation of *A. cinnamomea*. When no yeast extract powder was added (0 g/L), the biomass of *A. cinnamomea* is slightly declined but its sporulation is the highest and its sporulation ability is the strongest at more than 35 times that of the control. (3) Compared with nitrogen source, the different contents of glucose (carbon source) for continuous culture has a more significant effect on the biomass and also has significant effect on the sporulation capability of *A. cinnamomea*. The highest sporulation and strongest sporulation capacity of *A. cinnamomea* are found when no yeast extract powder is added and the content of glucose is 5.0 g/L.

When there is an absence of nitrogen source, *A. cinnamomea* can be rapidly induced to produce a large number of spores. A medium with rich nutrition exhibits an inhibiting effect on the asexual sporulation of *A. cinnamomea*. RNA-seq and transcriptomic analyses were then carried out for the following samples to explore the molecular regulatory mechanism underlying the asexual sporulation induced by nutrient limitation in the submerged fermentation of *A. cinnamomea*: (1) *A. cinnamomea* mycelium pellets cultured in fermentation medium for 9 days (recorded as “CK”); (2) *A. cinnamomea* mycelium pellets cultured in the nutrient-limiting medium containing 0 g/L yeast extract powder, 5.0 g/L glucose, 1.44 g/L KH_2_PO_4_, and 0.38 g/L MgSO_4_ at initial pH of 3.9 for another 4 days (recorded as “Poor”); and (3) *A. cinnamomea* mycelia cultured in rich nutrition medium containing 10.0 g/L yeast extract powder, 15.0 g/L glucose, 1.44 g/L KH_2_PO_4_, and 0.38 g/L MgSO_4_, at initial pH of 3.9 for another 4 days (recorded as “Rich”).

### 3.3. RNA-Seq and Statistical Analysis

#### 3.3.1. Statistical Analysis of Sample Repeatability and Differentially Expressed Genes

According to the FPKM values (Appendix A) of the unigene database, the principal component analysis and correlation analysis were performed (Figure 1) to investigate the consistency of the three biological repetitions and the distribution of differentially expressed genes in the three groups of samples.

Figure 1A shows that the distance among the three biological replicate samples in each group is centralized, and the Pearson correlation coefficients are more than 0.94. This finding indicates that the three biological replicate samples in each group have good repeatability. Meanwhile, the three groups are distinguishable, indicating significant difference among them (Figure 1B).

#### 3.3.2. GO Functional Analysis of Differentially Expressed Genes

Gene ontology (GO) functional classification enrichment analysis was carried out on the differentially expressed genes, and the results are shown in Figure 2.

Figure 2 shows that compared with that in the control group (“CK”), the expression of genes related to reproduction is significantly upregulated under nutrient limitation (“Poor”) but significantly downregulated under rich nutrition (“Rich”). Meanwhile, the expression of genes related to nutrient reservoir activity is significantly downregulated under nutrient limitation but significantly upregulated under rich nutrition. These genes with significant changes in expression and opposite changing trends in different nutritional conditions are of great value for further research. In addition, the genes involved in reproductive processes, signaling, receptor activity, and other biological processes show significantly changed expression under different nutritional conditions. With regard to functional classification, further bioinformatic analysis is required for these genes.

##### 3.3.3. iPath Analysis of Differentially Expressed Genes

Interactive pathway (iPath) analysis is an effective method to determine the relations or interactions among massive pathways. Here, the iPath 2.0 online analysis tool (http://pathways.embl.de accessed on 16 December 2021) was used for the iPath analysis of differentially expressed genes. After all differentially expressed genes in the three groups of samples were combined and compared in pairs, iPath analysis (Figure 3) was conducted on metabolic pathways (Appendix A) and regulatory pathways (Appendix A).

Figure 3 shows that the differentially expressed genes in the global overview of metabolic pathway are mainly focused on lipid metabolism, carbohydrate metabolism, amine acid metabolism, nucleotide metabolism, and energy metabolism. The differentially expressed genes in the global overview of regulatory pathway are mainly focused on translation, folding, sorting and degradation, and replication and repair. Many signal transduction pathways are involved in the regulation of the asexual sporulation of filamentous fungi [18,19].

The differentially expressed genes are involved in only one signal transduction pathway, that is, the two-component system (TCS). TCS widely exists in bacteria and fungi but has not been found in animals. It mainly consists of sensor histidine kinase and response regulator (RR). Among which, RR is usually a transcription factor [20]. The life activities of TCS include the absorption of nutrients (carbon, nitrogen, and phosphorus), response to physical or chemical stimuli (light, pH, and hypertonicity), and complex development processes (asexual sporulation, and fruiting body development) [20]. Therefore, the differentially expressed genes involved in TCS may be related to the regulation of the asexual sporulation of *A. cinnamomea.*

### 3.4. Bioinformatic Analysis

First, a local protein database (Appendix A) consisting of more than 10 thousand proteins related to the asexual sporulation of filamentous fungi was established. Another local protein database (Appendix A) consisting of 504 protein sequences encoded by the differentially expressed genes from GO functional analysis and iPath metabolic pathway analysis was established. BioEdit software was then employed to compare the two local databases, and 16 genes/proteins related to asexual sporulation of filamentous fungi are finally matched (Table 4).

Among the 16 genes shown in Table 4, *flbC*, *flbD*, *brlA*, *vosA*, and *pkaA* are involved in the central signal pathway of asexual sporulation mediated by FluG in *A. cinnamomea* [15]. *rco-3* gene encodes a glucose transporter, *tmpA* gene encodes a membrane oxidoreductase, *areA* gene encodes a regulator of nitrogen starvation response, *atg5* gene encodes an autophagy protein, *ech42* gene encodes an endochitinase, *gluC* gene encodes a glycoside hydrolase, *eng1* gene encodes an endo 1,3-beta glucanase, *cre1* gene encodes a carbon catabolite repressor, and *chsD* gene encodes a chitin synthase. Moreover, *CYP51A* gene encodes sterol 14a-demethylase enzymes, which mainly exist in the cytoplasm of mycelium and conidia of filamentous fungi. Knockdown of *CYP51A* gene can significantly reduce the sporulation and pathogenicity of *Magnaporthe oryzae*, indicating the involvement of this gene in the regulation of the asexual sporulation of *Magnaporthe grisea* [21].

Nutrient limitation is one of the most effective ways to induce the asexual sporulation of filamentous fungi, but the molecular regulatory mechanism remains unclear [22]. Researchers speculate about the role of some sensors for different nutrients in filamentous fungi. These sensors can transmit signals of different nutritional states in the environment to cells and then activate or promote the expression of related genes, thereby starting the related response mechanism [23]. Early in 1996, these nutrition sensors were reported in *Saccharomyces cerevisiae* by Ozcan et al. [24], who found that two glucose transporters (Rgt2p and Snf3p) act as glucose concentration sensors to generate corresponding intracellular signals and induce the expression of related genes in *S. cerevisiae*. In addition, the asexual sporulation of *Neurospora crassa* in submerged fermentation can be induced under the conditions of sufficient nutrients but with carbon limitation [25]. However, after the *outrco-3 gene* was knocked down, *N. crassa arco-3* knockout mutant still produced a large number of spores despite the presence of all nutrients (including carbon source). This finding showed that the transporter encoded by *rco-3* gene can regulate the asexual sporulation of *N. crassa* as a glucose sensor [25].

Asexual sporulation and exploratory hyphae formation are the two survival strategies of filamentous fungi under nutrient deficiency; however, both require a large amount of energy and building blocks [26]. Autophagy and autolysis are extreme and effective ways for filamentous fungi to quickly obtain nutrients and cell materials under nutrient deficiency. Extreme nutrient deficiency will activate or promote the massive expression of genes related to autophagy, autolysis, and asexual sporulation in filamentous fungal cells [27,28,29]. For example, in *Aspergillus nidulans*, carbon starvation will induce the synthesis of a large number of proteases to promote cell autolysis [30]. The two transcriptional activators encoded by *atgA* and *atgH* genes play an important role in autolysis and are highly expressed under carbon starvation or nitrogen starvation [27]. *atg5* is also one of the members of transcription activator families that promote cell autolysis [31]. In *Trichoderma atroviride*, *ech42* is highly expressed under carbon starvation, and the endochitinase encoded by *ech42* gene promotes the degradation of cell wall and further induces autolysis to provide a large number of building blocks for the synthesis of cell wall of newborn cells [32]. Chitin synthase plays an important role in the synthesis of spore cell wall [33]. Glycoside hydrolase is one of the key enzymes that degrade the cell wall during the autolysis of fungi and is found in *A. nidulans*. Its expression is significantly increased under carbon starvation to promote autolysis [26]. Carbohydrate active enzymes play an important modification role in the synthesis of the asexual spore cell wall of filamentous fungi by promoting spore maturation [33]. In submerged fermentation, *tmpA* gene is necessary for carbon starvation to induce the asexual sporulation of *A. nidulans*, and the knockout of *tmp A* gene leads to a significant decline in the spore production of *A. nidulans* [34]. *areA* gene encodes a transcription factor and greatly accumulates in the nucleus under carbon starvation; meanwhile, nitrogen starvation-responsive genes are activated to cope with the pressure of nitrogen starvation [35]. *cre1* gene encodes a C2H2 zinc finger structure transcription factor, which prompts organisms to prioritize the use of carbon sources with high nutritional value by inhibiting its expression via binding to the promoter region of the target gene. In *Trichoderma reesei*, *cre1* also inhibits the absorption of nitrogen source substances by bacteria [36].

### 3.5. RT-qPCR Analysis

RT-qPCR analysis (Figure 4) was carried out on the 18 genes related to the asexual sporulation signal pathway induced by FluG [15] and the remaining 11 genes in Table 4.

Figure 4 shows that among the genes related to the FluG-mediated signal pathway, most of the genes that inhibit sporulation (such as *sfgA*, *ganB*, *fadA*, *veA*, *velB*, and *vosA*) are downregulated under nutrient limitation. When nutrition is sufficient, their expression is slightly upregulated. On the contrary, most of the genes that promote sporulation (such as *flbB*, *flbC*, *flbD*, *brlA*, *abaA*, *wetA*, and *stuA*) are upregulated. When nutrition is sufficient, the changes are unclear. However, *nsdD*, *flbB*, *flbC*, *flbD*, *brlA*, *abaA*, and *wetA* show the most significant changes in their expression. Especially for *flbD* and *brlA*, the difference in the expression can reach up to 5–6 times. In addition, the genes promote the autophagy or autolysis of cells are significantly upregulated under nutrient limitation but downregulated when nutrition is sufficient. The changes in the expression of *rco-3*, *tmpA*, and *areA* under nutrient limitation are the most significant. Therefore, *flbD*, *brlA rco-3*, *tmpA*, and *areA* genes were believed to be critical in promoting the asexual sporulation of *A. cinnamomea* under nutrient limitation. *flbB* and *flbC* genes were also considered to play an important regulatory role. The upregulation of *abaA* and *wetA* genes may be caused by the regulation of their upstream genes *brlA* and *abaA*.

### 3.6. Model Diagram of A. cinnamomea Asexual Sporulation Signal Pathway Induced by Nutrition Limitation

Finally, the signal pathway of *A. cinnamomea* asexual sporulation induced by nutrient limitation and mediated by FluG was predicted (Figure 5).

Carbon or nitrogen starvation in *A. nidulans* induces the expression of the *brlA* gene in different degrees [37]. Under carbon starvation, the expression of *brlA* gene increases rapidly and largely. In submerged fermentation, the *tmpA* gene is necessary for nitrogen starvation to induce the asexual sporulation of *A. nidulans* and is strongly induced by nitrogen starvation signal; the knockout of *tmpA* gene leads to the significant downregulation of *brlA* gene and the significant decrease in the sporulation of *A. Nidulans* [34]. In addition, *tmpA* and *fluG* can regulate the asexual sporulation of *A. nidulans* through two different ways of gene knockout and overexpression [34]. In *A. nidulans*, the knockout of any one of the *flbA*, *flbB*, *flbC*, *flbD*, *fluG*, and *tmpA* genes will lead to the same phenotype and spore production will be significantly reduced. These findings indicate that *tmpA* has similar functions to *flbA*, *flbB*, *flbC*, and *flbD* as upstream regulators of *brlA* gene [38]. In addition, *flbD* is also involved in the response to nitrogen starvation, and its expression is significantly increased under carbon starvation [38]. On the basis of the above bioinformatic and RT-qPCR analysis results and relevant references [18,19,23,24,25,28,31,37,38,39,40,41,42], the molecular regulatory mechanism underlying the asexual sporulation of *A. cinnamomea* induced by nutrient limitation was predicted (Figure 5).

The regulatory signal pathway of *A. cinnamomea* asexual sporulation induced by nutrient limitation is as follows: under nitrogen starvation, the corresponding sensors transmit signals to genes, such as *areA* and *tmpA*, and promote their expression. AreA has a direct or indirect effect on *flbD* and promotes its expression, thereby enhancing the expression of *brlA*. Meanwhile, TmpA has a direct or indirect effect on *brlA* and promotes its expression. Under carbon starvation, the transporter Rco-3 act as a glucose sensor that directly or indirectly transmits the signal to *brl**A* and promote its expression. BrlA promotes the expression of *abaA* gene, which further enhances the expression of *wetA* gene. *wetA* gene then directly leads to asexual sporulation and promotes spore maturation. Under nutrition (carbon or nitrogen) starvation, the corresponding sensors directly or indirectly transmit signals to genes, such as *gulC* and *chsD*, and promote their expression, thus benefitting the synthesis of spore cell wall components and enhancing spore maturation. *gulC* can also promote cell autolysis, which provides energy and building blocks for sporulation.

## 4. Conclusions

In our previous study, the central signaling pathway of asexual sporulation (*fluG* → *flbB*/*flbC*/*flbD* → *brlA* → *abaA* → *wetA* → sporulation) in *A. cinnamomea* duiring submerged fermentation was revealed. In the present study, comparative transcriptomics was used to reveal the regulatory mechanism underlying asexual sporulation of *A. cinnamomea* induced by nutrient limitation in submerged fermentation. It was found that the signals of nutrient limitation (including carbon and nitrogen starvation) is firstly responded to by the corresponding sensors, such as Rco-3. Then, the sensors cause an effect on the downstream genes, such as *areA*, *tmpA*, *gluC*, and *chsD*. These genes play a direct or indirect role in enhancing the expression of the genes in the central signaling pathway, such as *flbD*, *brlA*, and *wetA*, thereby promoting the sporulation. This study provides a guidance for constructing *A. cinnamomea* strains with excellent sporulation performance. For example, in order to obtain an *A. cinnamomea* strain that can produce abundant spores, even under nutrient-rich conditions, it is possible to overexpress *rco-3* and/or *tmpA* genes. In this way, both biomass and sporulation can be taken into account, thereby greatly improving the sporulation efficiency of *A. cinnamomea* in submerged fermentation. Accordingly, the preparation efficiency of inoculum for submerged fermentation is markedly improved.

## Figures and Tables

**Figure 1 foods-11-02715-f001:**
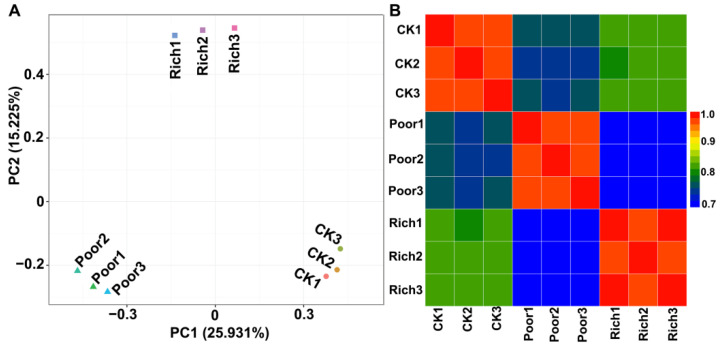
Statistical analysis of sample repeatability and differentially expressed genes. (**A**): PCA analysis; (**B**): correlation analysis.

**Figure 2 foods-11-02715-f002:**
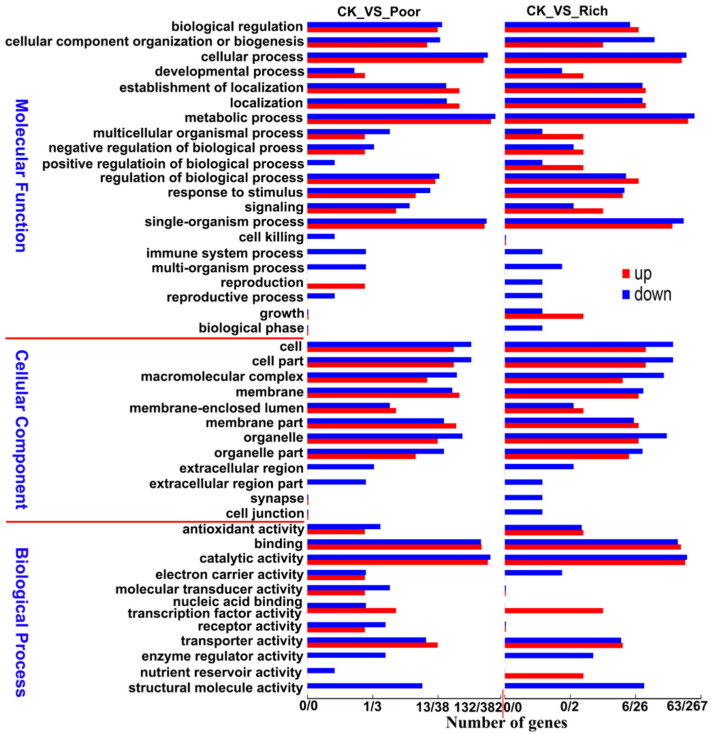
GO classification enrichment analysis of the differentially expressed genes.

**Figure 3 foods-11-02715-f003:**
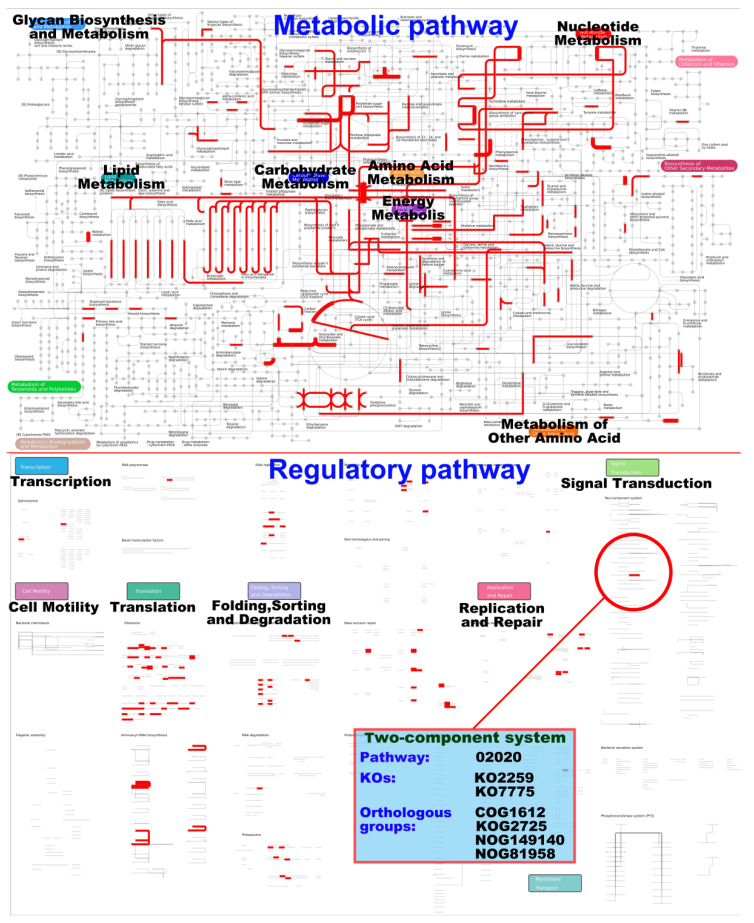
iPath analysis of the differentially expressed genes.

**Figure 4 foods-11-02715-f004:**
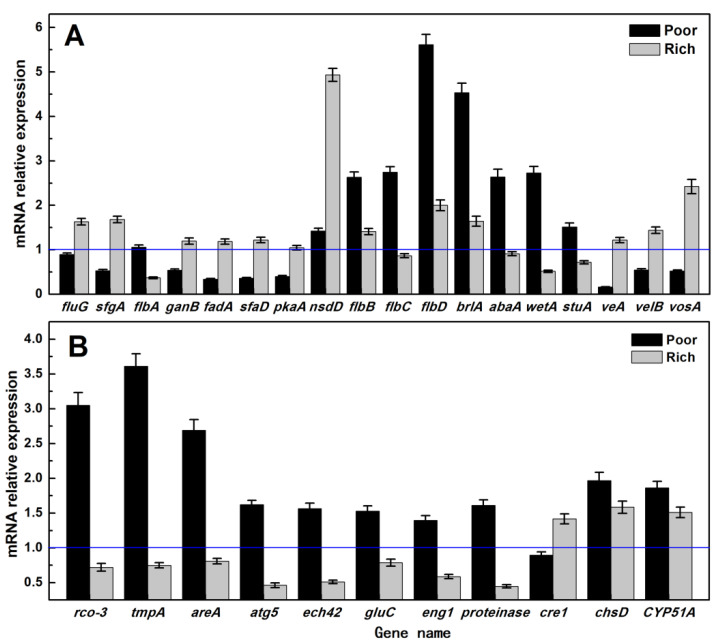
Different nutritional conditions effect on the genes involving in condition of *A*. *cinnamomea*. (**A**): changes in the expression of genes related to asexual sporulation signal pathway mediated by *A. cinnamomea* FluG; (**B**): changes in the expression of other genes related to asexual sporulation of *A. cinnamomea* matched with the local protein database; *A. cinnamomea* 18S rRNA gene was taken as the internal reference, and mycelium of control group was taken as CK.

**Figure 5 foods-11-02715-f005:**
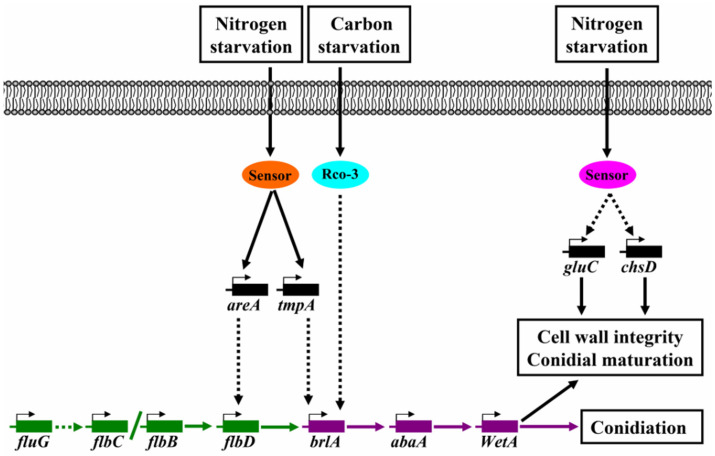
Proposed nutrient limitation induced signaling pathway of conidiation for *A*. *cinnamomea*.

**Table 1 foods-11-02715-t001:** Primers used for RT-qPCR.

Gene Name	Upstream Primer (5′→3′)	Downstream Primer (5′→3′)	Product (bp)
*rco3*	GCCGCCGACTCCCTCTG	AATGAACAAGCAAGCACTGACG	140
*tmpA*	GGCAGAAATTCCAAGAGCATAGTC	CACCTCAGGCACAACCATCC	178
*areA*	GTAGAGTGAGGCAAGGCAGATG	TGTCCAATTCAGTCCGCATACC	139
*atg5*	GGCGAGGAGGTATAATATGAGTGG	GGTGACGAAGGAGCGAAAGC	152
*ech42*	TCGCTGTATTTCTGTCGCTCAC	GCACCAACACCGCTCTACC	132
*gluC*	CAGGCGAGATGGTAAGGATACG	AGCGGGAAATGATGGTGAGC	141
*eng1*	GAGGAGGATTGGATGCGGATG	AACGAGGTCTCTAACAGTGATGC	161
*proteinase*	AGAAAGGGTGGTCAAGGTTATGC	CAGAAGAGGAGGTGCGAGATTAC	175
*cre1*	TCTCTTCGCTTGGTTCACATCC	CGCTTGTATATGGCTCCTGGTC	180
*chsD*	ATATGGTATGTGGAGTGGTTGGC	GGCGGTGGCGATGATTGG	175
*CYP51A*	GTGTTGGAAATGTTGACGAGGAC	GCGGAGTGGAAGATGACAAGG	178
18S rRNA	GCTGGTCGCTGGCTTCTTAG	CGCTGGCTCTGTCAGTGTAG	123

**Table 2 foods-11-02715-t002:** Composition analysis of the fermentation broth of *A. cinnamomea*.

Component	Content (g/L)	
	0 Day	5 Days
glucose	20.1 ± 0.2	15.1 ± 1.4
nitrogen	1.2 ± 0.1	0.6 ± 0.0
SO_4_^2−^	1.2 ± 0.0	0.3 ± 0.0
PO_4_^3−^	2.1 ± 0.1	1.0 ± 0.1
pH	4.5 ± 0.0	3.9 ± 0.1

Note: the medium contained 20.0 g/L glucose, 10.0 g/L yeast extract powder, 3.0 g/L KH_2_PO_4_, and 1.5 g/L MgSO_4_ with initial pH of 4.5, then cultured at 26 °C and 150 r/min for 10 days with spore inoculum of 1.0 × 10^6^ spores/mL; “0 day”: the fresh medium that did not begin to culture; “5 days”: the fermentation broth cultured for 5 days.

**Table 3 foods-11-02715-t003:** Effect of different nitrogen and carbon contents on the conidiation of *A. cinnamomea*.

Content (g/L)	Biomass (g/L)	Sporulation (×10^5^ Spores/mL)	Sporulation Capability (×10^8^ Spores/g)	Residual Sugar (g/L)
Nitrogen (yeast extract powder) contents (g/L)				
0	3.6 ± 0.3 ^b^	230.0 ± 20.3 ^f^	63.5 ± 2.7 ^f^	4.9 ± 0.1 ^a^
0.5	4.0 ± 0.3 ^bc^	63.5 ± 4.5 ^e^	15.8 ± 1.0 ^e^	4.7 ± 0.1 ^a^
1.0	4.0 ± 0.3 ^bc^	35.5 ± 3.3 ^d^	8.9 ± 0.6 ^d^	4.7 ± 0.1 ^a^
5.0	4.6 ± 0.3 ^cd^	8.3 ± 0.3 ^c^	1.8 ± 0.1 ^c^	4.6 ± 0.1 ^a^
10	4.9 ± 0.4 ^d^	4.0 ± 0.5 ^b^	0.8 ± 0.0 ^b^	4.6 ± 0.2 ^a^
Control				
CK1	4.8 ± 0.4 ^d^	2.8 ± 0.3 ^b^	0.6 ± 0.0 ^b^	4.5 ± 0.2 ^a^
CK2	4.7 ± 0.4 ^d^	2.3 ± 0.3 ^b^	0.5 ± 0.0 ^b^	4.5 ± 0.1 ^a^
5 days	1.5 ± 0.1 ^a^	0.0 ± 0.0 ^a^	0.0 ± 0.0 ^a^	13.5 ± 1.2 ^b^
Carbon (glucose) contents (g/L)				
0	1.3 ± 0.1 ^a^	165.0 ± 12.5 ^d^	132.0 ± 8.7 ^e^	0.1 ± 0.0 ^a^
2.0	1.9 ± 0.2 ^b^	262.5 ± 25.2 ^e^	135.3 ± 9.1 ^e^	0.1 ± 0.0 ^a^
5.0	2.8 ± 0.3 ^c^	487.5 ± 30.5 ^f^	176.6 ± 9.7 ^f^	0.1 ± 0.0 ^a^
10	3.0 ± 0.1 ^c^	256.3 ± 13.7 ^e^	84.6 ± 4.2 ^d^	0.2 ± 0.0 ^a^
20	3.9 ± 0.3 ^d^	114.3 ± 8.0 ^c^	29.1 ± 2.0 ^c^	8.9 ± 0.8 ^c^
Control				
CK1	4.4 ± 0.2 ^e^	4.3 ± 0.3 ^b^	0.9 ± 0.0 ^b^	4.6 ± 0.1 ^b^
CK2	4.3 ± 0.3 ^e^	4.5 ± 0.2 ^b^	1.0 ± 0.0 ^b^	4.5 ± 0.2 ^b^
5 days	1.8 ± 0.1 ^b^	0.0 ± 0.0 ^a^	0.0 ± 0.0 ^a^	15.1 ± 1.4 ^d^

Note: For all the groups, *A. cinnamomea* was first cultured in fermentation medium for 5 days. The broth was allowed to stand for 5 min, and the supernatant was then removed with a pipette under sterile conditions. The obtained mycelium pellets were washed with normal saline three times and then treated as follows. (1) For nitrogen: added with 100 mL of medium containing 0, 0.5, 1.0, 5.0, or 10.0 g/L yeast extract powder, 15.0 g/L glucose, 1.44 g/L KH_2_PO_4_, and 0.38 g/L MgSO_4_ with initial pH of 3.9, and continued to culture for 4 days. (2) For carbon: added with 100 mL of medium containing 0, 2.0, 5.0, 10.0, or 20.0 g/L glucose, 0 g/L yeast extract powder, 1.44 g/L KH_2_PO_4_, and 0.38 g/L MgSO_4_ with initial pH of 3.9, and continued to culture for 4 days. (3) For control: “CK1”: *A. cinnamomea* was cultured in fermentation medium for 9 days without any treatment; “CK2”: mycelium pellets were washed with normal saline three times, then transferred back the supernatant, and continuously cultured for another 4 days; “5 days”: the fermentation broth of *A. cinnamomea* cultured for 5 days in fermentation medium; (4) “Sporulation capability” means the sporulation produced by per gram of mycelium, and was calculated by dividing the sporulation by the biomass; (5) Different letters in the same column of the table indicate that the difference is significant at the level of 0.05.

**Table 4 foods-11-02715-t004:** Differential expression genes with possible involvement in the condition of *A. cinnamomea*.

Unigene ID	Genome ID	Gene Name	Accession Number	E Value	Homology (%)	Coverage (%)
c6157_g2	ACg005466	*flbC*	OAS999571	3.40 × 10^−10^	47.458	8
c9703_g1	ACg001829	*flbD*	EEQ33126.1	7.23 × 10^−25^	43.802	31
c5765_g1	ACg000119	*brlA*	AAM95989.1	5.33 × 10^−11^	53.846	12
c6072_g2	ACg005882	*vosA*	XP_0095505191	5.54 × 10^−59^	39.61	88
c2971_g1	ACg005363	*pkaA*	EFL410741	6.84 × 10^−29^	29.032	65
c6203_g1	ACg005139	*rco-3*	OBZ74137.1	5.25 × 10^−106^	36.154	94
c5652_g1	ACg002449	*tmpA*	AAP13095.2	3.48 × 10^−99^	38.318	88
c6469_g1	ACg003505	*areA*	CCO35477.1	6.08 × 10^−32^	37.908	25
c7055_g1	ACg003525	*atg5*	KYQ43884.1	2.56 × 10^−180^	71.023	97
c4436_g1	ACg008078	*ech42*	ABS82797.1	6.74 × 10^−28^	26.879	70
c6333_g2	ACg007557	*gluC*	EHK46811.1	7.11 × 10^−142^	47.992	32
c5534_g2	ACg005514	*eng1*	AHI42991.1	6.44 × 10^−149^	62.776	99
c6649_g1	ACg000307	*proteasome*	CEL57564.1	5.21 × 10^−157^	82.609	99
c5803_g1	ACg008021	*cre1*	AAT34979.1	4.24 × 10^−14^	51.563	14
c4252_g1	ACg006463	*chsD*	CDM33077.1	2.03 × 10^−14^	21.918	26
c6977_g1	-	*CYP51A*	AIF79427.1	5.06 × 10^−10^	22.68	44

Note: “Unigene ID” is the code of unigene generated in the process of software assembly; “Genome ID” is the code corresponded by the gene matched with Unigene in *A. cinnamomea* genome; “-“ means the gene unmatched in *A. cinnamomea* genome database; “Accession number” is the number in NCBI of the protein matched with Unigene in local protein database; “E value”, “Homology”, and “Coverage” are used to describe the matching between Unigene and the corresponding protein in the local protein database. If E value is lower, Homology and Coverage are higher, the matching degree is higher. If E value ≤ 10^−6^, it means that the matching is successful.

## Data Availability

Data are contained within the article or Appendix A and available upon request from the corresponding author.

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
