# Peer review of "Comparative Transcriptomic Analyses Reveal the Regulatory Mechanism of Nutrient Limitation-Induced Sporulation of Antrodia cinnamomea in Submerged Fermentation"

_foods, 2022, doi:10.3390/foods11172715_

Round 1
Reviewer 1 Report
The manuscript is reviewed critically.
Methodology: This section needs to add more clarification and details.
Statistical analysis: Page 4 and 5: The use of PCA and correlational analysis needs to be highlighted here.
Results: Figure 1C is somewhat confusing. I do not understand what tests have been conducted to produce the sub-cell as shown. Also, results and discussion were presented from page 5 to 14. This is a very long section.
Conclusion: This section is somewhat simple. The writing is similar to the one shown in the abstract and last paragraph of discussion. Basically the content is repeating what have been written previously. Also, the last sentence is considered meaningless and has no drive for future directions.
Author Response
Response for Review 1
Q1. Methodology: This section needs to add more clarification and details.
A: Thank you so much for your kindly suggestion. I have added more details in the “Materials and Methods” section.
Q2. Statistical analysis: Page 4 and 5: The use of PCA and correlational analysis needs to be highlighted here.
A: Thank you so much for your kindly suggestion. I added the use of PCA and correlational analysis in the section of “2.7. RNA-seq and bioinformatic analysis”.
Q3. Results: Figure 1C is somewhat confusing. I do not understand what tests have been conducted to produce the sub-cell as shown.
A: Thank you so much for your kindly remind. Figure 1C really doesn't make much sense, so I removed it.
Q4. Also, results and discussion were presented from page 5 to 14. This is a very long section.
A: Thank you so much for your kindly remind. I have simplified the “results and discussion” section and deleted lots of sentences.
Q5. Conclusion: This section is somewhat simple. The writing is similar to the one shown in the abstract and last paragraph of discussion. Basically the content is repeating what have been written previously. Also, the last sentence is considered meaningless and has no drive for future directions.
A: Thank you so much for your professional comment. I have rewritten the conclusion section.

Reviewer 2 Report
The manuscript “Comparative transcriptomic analyses reveals the regulatory mechanism of nutrient limitation-induced sporulation of Antrodia cinnamomea in submerged fermentation” is well written. This can be considered worth publishing in journal “Foods” after following modifications:
1. The objective of the study should be presented in a more clear way.
2. A schematic diagram of experimental setup must be provided foe better understanding.
3. A separate abbreviation should be provided.
4. Section 2. Instrument detail should be written as equipment name (model, company, country)
5. Author can discuss the comparison of their study with other related research.
6. Section 2.5 Authors discussed the effect of only 2 conditions (i-e contents of glucose and contents of yeast extract powder) on sporulation of A. cinnamomea. There are other factors affecting, which will be discussed.
7. Section 2.2, 2.3 and 2.4. The authors adopted the methods from another research, or it is their own. If not, then provide appropriate reference.
8. Figure 1. X-axis and y-axis description is missing.
9. The diagram of mechanism of asexual sporulation in the submerged fermentation of A. cinnamomea should be given with more detail.
Author Response
Response for Review 2
The manuscript “Comparative transcriptomic analyses reveals the regulatory mechanism of nutrient limitation-induced sporulation of Antrodia cinnamomea in submerged fermentation” is well written. This can be considered worth publishing in journal “Foods” after following modifications:
Q1. The objective of the study should be presented in a more clear way.
A: Thank you so much for your professional suggestion. I have revised the description to highlight the objective of this study in the “abstract”, “introduction” and “conclusion” sections.
Q2. A schematic diagram of experimental setup must be provided foe better understanding.
A: Thank you so much for your kindly remind. I supplemented a “Graph Abstract” as follow, and it will publish with the paper.
Q3. A separate abbreviation should be provided.
A: Thank you so much for your kindly remind. I have added a separate abbreviation after the “conclusion” section.
Q4. Section 2. Instrument detail should be written as equipment name (model, company, country)
A: Thank you so much for your professional suggestion. I have revised the instrument detail as equipment name (model, company, country).
Q5. Author can discuss the comparison of their study with other related research.
A: Thank you so much for your professional suggestion. In fact, so far almost only our team conducts the research on asexual sporulation of A. cinnamomea by submerged fermentation. In 2013, we discovered the phenomenon of asexual sporulation by submerged fermentation of A. cinnamomea and reported it for the first time in Applied Microbiology and Biotechnology (Gen et.al., 2013); In 2014, the arthrospores of A. cinnamomea from submerged fermentation were firstly used as inoculum, and an efficient submerged fermentation process of A. cinnamomea based on spore inoculation was established (Lu et.al., 2014). In 2015, we further optimized the previous fermentation conditions and established a repeated batch fermentation process of A. cinnamomea based on spore inoculation, which greatly improved the production efficiency and benefits as the additional preparation of arthrospores is not need (Li et.al., 2015). In 2017, we combined comparative proteomics and transcriptomics analysis to reveal the FluG-mediated signaling pathway underlying asexual sporulation of A. cinnamomea during submerged fermentation for the first time (Figure 1) (Li et.al., 2017).
Figure 1. Proposed FluG-mediated signaling pathway might relate to the asexual sporulation of A. cinnamomea. UDA: upstream developmental activation; CDP: central developmental pathway. (Li et.al., 2017)
The present study is actually a further supplement to our previous study in 2017. Thus, our previous related studies were cited and compared in this manuscript. In addition, this study also referenced and compared other related researchs on nutrient starvation-induced sporulation in filamentous fungi, such as reference of [18], [25], [37], [41], and [43] in the manuscript. Thanks again for your kindly remind.
Q6. Section 2.5 Authors discussed the effect of only 2 conditions (i-e contents of glucose and contents of yeast extract powder) on sporulation of A. cinnamomea. There are other factors affecting, which will be discussed.
A: Thank you so much for your professional suggestion. We actually systematically investigated lots of conditions that effect on sporulation of A. cinnamomea in our previous studies. In 2013, we studied the effect of carbon sources (including citric acid, D-fructose, D-galactose, D-glucose, D-mannose, mannitol, soluble starch, succinate, and sucrose), nitrogen sources (including beef extract, casein amino acid, L-glutamic acid, L-valine, peptone, sodium nitrate, soy flour, tryptone, and yeast extract powder), C/N ratio (including 1:1, 2:1, 20:1, 40:1, and 60:1) , content of MgSO4 (including 0.2, 0.5, 1.0, 1.5, and 2.0 g/L), content of KH2PO4 (including 0.6, 1.0, 2.0, 3.0, and 4.0 g/L), initial pH (including 3.0, 4.0, 5.0, 6.0, 7.0, 8.0, and 9.0), and inoculum size (including 0.5, 1.0, 1.5, 2.0, and 2.5×105 spores/mL) on sporulation of A. cinnamomea in submerged fermentation (Geng et.al., 2013). In 2015, we compared the effect of peptone contet (including 0.5, 1.0, 1.5, and 2.0 g/L) and inoculum size (including 0.1, 0.5, 1.0, 5.0, and 10.0 ×106 spores/mL) on sporulation of A. cinnamomea in submerged fermentation (Li et.al., 2015). In 2022, we studied the effect of metal ions (including Na+, Mg2+, Cu2+, Al3+, Ca2+, Fe2+, and Zn2+) on sporulation of A. cinnamomea in submerged fermentation (Li et.al., 2022).
However, in the present study, the effect of different conditions on sporulation of A. cinnamomea is not the focus of our research. We here discussed the effects of carbon and nitrogen sources on sporulation of A. cinnamomea just to obtain the mycelium samples cultured under different nutrient conditions with significant differences in sporulation. Then the mycelium samples were used for comparative transcriptomic analysis and ultimately to reveal the regulatory mechanism underlying asexual sporulation of A. cinnamomea induced by nutrient limitation in submerged fermentation. That is why we only discussed the effect of contents of glucose and yeast extract powder on sporulation of A. cinnamomea in the present study.
Q7. Section 2.2, 2.3 and 2.4. The authors adopted the methods from another research, or it is their own. If not, then provide appropriate reference.
A: Thank you so much for your kindly remind. In fact, most methods are from our previous studies, and I provided the references to them. The ones without reference are used only in the present study.
Q8. Figure 1. X-axis and y-axis description is missing.
A: Thank you so much for your kindly remind. I have added the X-axis and y-axis in the Figure 1.
Q9. The diagram of mechanism of asexual sporulation in the submerged fermentation of A. cinnamomea should be given with more detail.
A: Thank you so much for your professional suggestion. As mentioned above, we used comparative proteomics and transcriptomics analysis to firstly reveal the FluG-mediated asexual sporulation signaling pathway underlying asexual sporulation of A. cinnamomea during submerged fermentation (Figure 1) (Li et.al., 2017). As the central signaling pathway of sporulation, it detailly demonstrated in the regulatory mechanism of the asexual sporulation process of A. cinnamomea in submerged fermentation. In this pathway, fluG, flbA, flbB, flbC, and flbD are the most important upstream genes; brlA , abaA, and wetA are the three core and essential genes that lead to the final asexual sporulation of A. cinnamomea. However, the study in 2017 did not explain why A. cinnamomea produces spores in the late stage of the submerged fermentation. Now, we know that it is because the nutrients is deficient at the later stage of fermentation, thereby the nutrients can not maintain the continued growth of A. cinnamomea. Then, the spors of A. cinnamomea are produced to survive. However, how did the nutrition limitation linduce the asexual sporulation of A. cinnamomea? This is the main content and objective of the present study. Therefore, this study is actually a further supplement and improvement to our study in 2017.
In fact, we also found that Ca2+ can significantly promote asexual sporulation of A. cinnamomea, and we used the comparative proteomics analysis to reveal the regulatory mechanism of Ca2+-induced asexual sporulation of A. cinnamomea in submerged fermentation (Figure 2) (Li et.al. , 2017). Similar to Figure 2, the main purpose of the Figure 5 herein is to explain how nutrient limitation signals are transmited to the FluG-mediated central signaling pathway. In order to obtain the diagram of Figure 5, I summarized more than 27 relevant references (including references of [18-44]) and combined the data from comparative transcriptomic analysis. Obviously, there may be more genes involved in signaling the nutrient deprivation. However, I searched and studied lots of references and really could not find any more relevant genes. I am so sorry. Nevertheless, the present study still has a high academic value and reference significance. Therefore, please kindly give your tolerance and understanding. Thank you very much!
Figure 2. Proposed model for Ca2+/calmodulin- and FluG-mediated submerged conidiation in A. cinnamomea. (Li et.al., 2017)
Reference
- Geng, Y.; He, Z.; Lu, Z.M.; Xu, H.Y.; Xu, G.H.; Shi, J.S.; Xu, Z.H. Antrodia camphorata ATCC 200183 sporulates asexually in submerged culture. Appl. Microbiol. Biot. 2013, 97(7), 2851-2858. https://doi.org/DOI 10.1007/s00253-012-4513-2.
- Lu, Z.M.; He, Z.; Li, H.X.; Gong, J.S.; Geng, Y.; Xu, H.Y.; Xu, G.H.; Shi, J.S.; Xu, Z.H. Modified arthroconidial inoculation method for the efficient fermentation of Antrodia camphorata ATCC 200183. Biochem. Eng. J. 2014, 87, 41-49. https://doi.org/10.1016/j.bej.2014.03.020.
- Li, H.X.; Lu, Z.M.; Geng, Y.; Gong, J.S.; Zhang, X.J.; Shi, J.S.; Xu, Z.H.; Ma, Y.H. Efficient production of bioactive metabolites from Antrodia camphorata ATCC 200183 by asexual reproduction-based repeated batch fermentation. Bioresource Techno. 2015, 194, 334-343. https://doi.org/10.1016/j.biortech.2015.06.144.
- Li, H.X.; Lu, Z.M.; Zhu, Q.; Gong, J.S.; Geng, Y.; Shi, J.S.; Xu, Z.H.; Ma, Y.H. Comparative transcriptomic and proteomic analyses reveal a flug-mediated signaling pathway relating to asexual sporulation of Antrodia camphorata. Proteomics 2017, 17, 1700256. https://doi.org/10.1002/pmic.201700256.
- Li, H.X.; Lu, Z.M.; Zhu, Q.; Geng, Y.; Shi, J.S.; Xu, Z.H.; Ma, Y.H. Effect of calcium on sporulation of Taiwanofungus camphoratus in submerged fermentation. Chinese Journal of Biotechnology 2017, 33(7): 1124−1135. https://doi.org/10.13345/j.cjb.170001.
- Li H.X.; Wang J.J.; Shi Y.; Ji D.; Gao Y.J.; Jia L.Q.; Rao S.Q.; Yang Z.Q. Metal ions promoting sporulation of Antrodia cinnamomea in submerged fermentation. Food and Fermentation Industries. 2022.

Round 2
Reviewer 2 Report
The authors revised accordingly